# SoftHash: High-dimensional Hashing with A Soft Winner-Take-All Mechanism

## Abstract

Locality-Sensitive Hashing (LSH) is a classical algorithm that aims to hash similar data points into the same bucket with high probability. Inspired by the fly olfactory system, one variant of the LSH algorithm called *FlyHash*, assigns hash codes into a high-dimensional space, showing great performance for similarity search. However, the semantic representation capability of *FlyHash* is not yet satisfactory, since it is a data-independent hashing algorithm, where the projection space is constructed randomly, rather than adapted to the input data manifold. In this paper, we propose a data-dependent hashing algorithm named *SoftHash*. In particular, *SoftHash* is motivated by the bio-nervous system that maps the input sensory signals into a high-dimensional space, to improve the semantic representation of hash codes. We learn the hashing projection function using a Hebbian-like learning rule coupled with the idea of Winner-Take-All (WTA). Specifically, the synaptic weights are updated solely based on the activities of pre- and post-synaptic neurons. Unlike the previous works that adopt the hard WTA rule, we introduce a soft WTA rule, whereby the non-winning neurons are not fully suppressed in the learning process. This allows weakly correlated data to have a chance to be learned to generate more representative hash codes. We conduct extensive experiments on six real-world datasets for tasks including image retrieval and word similarity search. The experimental results demonstrate that our method significantly outperforms these baselines in terms of data similarity search accuracy and speed.

## 1 Introduction

Locality Sensitive Hashing (LSH) (Indyk & Motwani, 1998) has become a widely studied technique in computer science, which projects input features into binary codes, helping to lessen computational time in similarity search. The objective of LSH is to cluster similar samples closely while separating dissimilar ones in the hash space. Classical LSH methods (Kang et al., 2016; Chen et al., 2020) are usually used to convert high-dimensional features into low-dimensional binary codes for fast and efficient nearest-neighbor search. Recently, inspired by the fruit fly olfactory circuit depicted in Figure 1(a), researchers (Dasgupta et al., 2017) have proposed a bio-inspired LSH, named *FlyHash*. LSH achieves sparse feature representation by converting high-dimensional features into higher-dimensional binary codes. A learnable *FlyHash* version is *BioHash* (Ryali et al., 2020), which is able to update synaptic weights of hashing network based on the input data, demonstrating an improved performance over the classical *FlyHash*. Although *BioHash* is a biologically plausible unsupervised algorithm, its hard winner-take-all (WTA) mechanism only allows one winner neuron to be updated in each learning step which greatly limits its ability to catch semantic information of input data.

Winner-Take-All (WTA) (Rumelhart & Zipser, 1985), an important competition mechanism in recurrent neural networks (Wang et al., 2019a). It represents a computational principle utilized in computational models of neural networks, wherein neurons compete with each other to become activated. The WTA network is commonly used in computational models of the brain, particularly for distributed decision-making or action selection in the cortex. It can be used to develop feature selectivity through competition in simple recurrent networks. The classical form of WTA is hard WTA where only the neuron with the highest activation can be updated while other neurons keep unchanged in the learning process. Such WTA is so crude that the input signal may not be fully

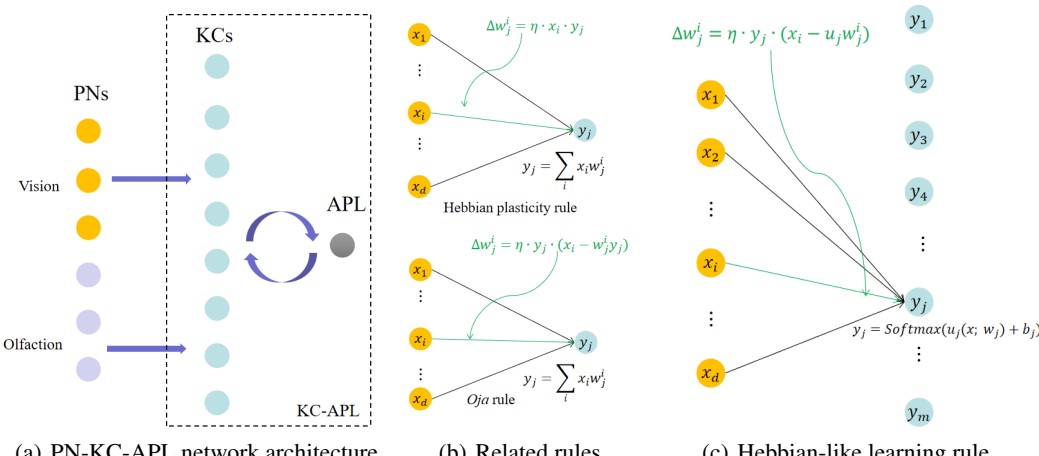

(a) PN-KC-APL network architecture  (b) Related rules  (c) Hebbian-like learning rule

Figure 1: (a) Schematic of the PN-KC-APL network architecture of the mushroom body (MB) in the fruit fly. MB is the main area of the Drosophila brain responsible for processing sensory information (Liang et al., 2021). It receives inputs from several sets of projection neurons (PNs) corresponding to several sensory modalities. The predominant one is olfaction (Bates et al., 2020), followed by vision (Li et al., 2020b) and thermo-hydro sensory system (Marin et al., 2020), etc. PNs are connected to approximately 2,000 Kenyon cells (KCs) through synaptic weights (Li et al., 2020a). KCs then provide feed-forwarded excitation to a single inhibitory neuron called anterior paired lateral (APL), and receive inhibitory feedback. Consequently, all but the top activated neurons are silenced in KCs. (b) Illustration of the Hebbian plasticity rule and *Oja* rule. (c) Illustration of the local Hebbian-like learning rule used in this work.

exploited by the neurons, affecting the semantic representation of the trained network. Compared to the hard WTA, the soft WTA allows more than one neuron to be updated with the input signal, which is more practical and effective in utilizing the input data to extract semantic features.

Based on the soft WTA, this paper introduces a hash function with learnable projection parameters, to improve the semantic representation of resulting hash codes. Building on inspiration from the biological evidence and the idea of dimension expansion, we design a novel hashing algorithm *SoftHash* (hashing with a soft winner-take-all mechanism) to generate sparse binary hash codes for the input data presented. *SoftHash* is a data-dependent hashing algorithm that learns synaptic weights and biases in a neurobiologically plausible way. We adopt a local Hebbian-like rule combined with the WTA mechanism to learn semantic features from input data. Especially, instead of making neurons compete via a hard WTA mechanism, we adopt a soft WTA mechanism, by which an exponential function is applied to the neurons, allowing weakly correlated data to be learned.

In this paper, we design a data-dependent hashing algorithm *SoftHash* to preserve the relative relationships between original data points through binary representations, enabling accurate and fast similarity search. The main contributions of this paper are summarized as follows:

- **Hashing algorithm:** We introduce *SoftHash*, a novel data-dependent hashing algorithm, which can produce sparse and yet discriminative high-dimensional hash codes.

- **Biologically plausible:** We employ a Hebbian-like local learning rule combined with the soft WTA mechanism to acquire semantic information from input data. Notably, this mechanism enables the learning of weakly correlated data, improving the semantic representation of produced hash codes.

- **Evaluation in different contexts:** Our extensive experiments on image retrieval tasks demonstrate significantly improved image retrieval precision over existing data-dependent hashing methods. Furthermore, on the word similarity search task, the performance of *Soft-Hash* produced binary hash code approaches original dense embeddings while taking less computation time.

## 2 RELATED WORK

### 2.1 HASHING ALGORITHMS AND SPARSE EXPANSIVE REPRESENTATIONS

In recent years, several data-dependent LSH methods have emerged, such as PCA hashing (Gong et al., 2012), spectral hashing (Weiss et al., 2008), graph hashing (Liu et al., 2014; Jiang & Li, 2015), and deep hashing (Su et al., 2018; Li & Li, 2022). These techniques aim to project high-dimensional features into low-dimensional binary representations. In contrast, *FlyHash* distinguishes itself from classical LSH algorithms in three key aspects: (1) it utilizes sparse, binary random weights; (2) it expands the dimensionality of the input data; and (3) it sparsifies the binarized output representations. *DenseFly* (Sharma & Navlakha, 2018) follows this direction to generate sparse expansive representations of the inputs by setting all indices with values $\geq 0$ to 1, and the remaining to 0, showing improved performance for nearest-neighbors retrieval. The random projection adopted by *FlyHash* and *DenseFly* is data-independent, however, the fruit flies "learn to hash" (Hige et al., 2015).

*Optimal Sparse Lifting* (Li et al., 2018) generates sparse binary output vectors for input data samples by approximately preserving pairwise similarities between input data samples. This is achieved by conducting constrained linear programming involving all training data in each learning step. However, in biological neural networks, the neuron responses are tuned by a synapse-change procedure that is physically local (Krotov & Hopfield, 2019). Motivated by this, *BioHash* (Ryali et al., 2020) applies local Hebbian-like rule to learn sparse expansive motifs, which demonstrate significantly better retrieval performance over LSH approaches (e.g., classical LSH, data-driven hashing, deep hashing) at short hash length. However, *BioHash* shows marginal improvements when the hash length reaches a certain level. This may be attributed to the hard WTA mechanism adopted by *Bio-Hash*, allowing only one neuron to become activated for any input sample. As will be explained soon, this hard WTA mechanism greatly limits the learning and expressive ability of the neural network in extracting useful hash codes.

### 2.2 HEBBIAN-LIKE RULES AND WTA MECHANISMS

Hebbian plasticity is a local correlation-based learning rule (Zhou, 2022) (Figure 1 (b)), which follows the principle that the synaptic change should only depend on the pre and post synaptic neurons' firing activities. The general form of the Hebbian plasticity rule can be described as follows:

$$\Delta w_j^i = \eta x_i y_j. \tag{1}$$

where $w_j^i$ is the synaptic weight connecting $i$th input to neuron $j$, $y_j = \sum_i x_i w_j^i$ is the output of neuron $j$. It is obvious that synaptic weights might grow unboundedly. Thus, constraints are typically introduced into the Hebbian plasticity rule to address this issue. One notable example is the *Oja* rule (Oja, 1982) (Figure 1 (b)), which is defined as follows:

$$\Delta w_j^i = \eta y_j (x_i - w_j^i y_j). \tag{2}$$

Theoretical analysis shows that the neuron, after being trained with *Oja* rule, tends to extract the principal component from a stationary input vector sequence (Oja, 1982).

The winner-take-all (WTA) mechanism (Rumelhart & Zipser, 1985) is primitive for neural computation, where neurons within a WTA neural circuit will compete to represent an input stimulus, and the winning one, with the closest receptive field, will suppress the rest. When coupled with the WTA mechanism, the Hebbian learning rule allows neurons to learn more discriminative feature representation. Notably, it has been shown that the Hebbian plasticity combined with hard WTA can approach the performance of error backpropagation algorithm in training shallow feedforward networks (Krotov & Hopfield, 2019). Subsequent studies have successfully applied this approach to other network structures (Grinberg et al., 2019; Gupta et al., 2021) as well as deeper networks (Shinozaki, 2020; Journé et al., 2022).

In this work, we extend these earlier studies by exploring the use of the Hebbian learning rule and soft WTA mechanism in learning discriminative hash codes, which demonstrate superior capability in capturing the weak correlations among data samples.

## 3 APPROACH

### 3.1 LEARNING WITH A SOFT WTA MECHANISM

In this section, we provide a comprehensive description of the proposed hashing algorithm, *Soft-Hash*. Our objective with *SoftHash* is twofold: to enable fast similarity search and to preserve the relative similarity between the original data points as much as possible.

Mathematically, the objective of the proposed hashing algorithm is to cluster the input data samples into some buckets. Let us denote an input data sample $x$ in a $d$-dimensional space as $x \in \mathbb{R}^d$ and its hash codes in a $m$-dimensional space as $h(x) \in \mathbb{R}^m$, where $h(\cdot)$ is the hash function. We further denote the weight matrix for synaptic connections as $W \in \mathbb{R}^{m \times d}$ and neuron bias as $b \in \mathbb{R}^m$.

We utilize a Hebbian-like rule (Moraitis et al., 2022) that combines a Bayesian theorem incorporating a soft-winner-takes-all (soft-WTA) mechanism with softmax smoothing. A summary of its probabilistic interpretation and mathematical description is presented in Table 1, wherein $q_j(\cdot)$ is the probability of neuron $j$.

Table 1: Summary of probability interpretations and mathematical descriptions.

| Probabilistic interpretation | Mathematical description |
|---|---|
| prior probability | $Q(C_j; b_j) = e^{b_j}$ |
| $j$-th component of mixture likelihood function | $q_j(x\|C_j; w_j) = e^{u_j(x; w_j)}$ |
| posterior probability | $q_j(C_j\|x; b_j) = y_j = \frac{e^{u_j + b_j}}{\sum_{l=1}^m e^{u_l + b_l}}$ |

The plasticity rule for any neuron $j$ can be written as

$$\tau \frac{dw_j^i}{dt} = \Delta w_j^i = \eta \cdot y_j \cdot (x_i - u_j w_j^i) \tag{3}$$

where $\tau$ is time constant of the plasticity dynamics, $w_j^i$ is synaptic weight from the $i$th neuron at the input layer to the $j$th neuron at the output layer, and $\eta$ is the learning rate. $u_j$ is a postsynaptic variable obtained by the inner product $\langle x, w_j \rangle = \sum_i w_j^i x_i$, where $w_j \in W$. $b_j$ is the bias term of neuron $j$, representing prior information stored by the neuron. Inspired by the spike-based learning rule (Nessler et al., 2013), the biases $b_j$ can be iteratively updated as per

$$\Delta b_j = \eta e^{-b_j}(y_j - e^{b_j}). \tag{4}$$

Based on the above parameters, the Bayesian posterior can be calculated by the following softmax function

$$y_j = \frac{e^{u_j + b_j}}{\sum_{l=1}^m e^{u_l + b_l}} \tag{5}$$

where $u_j + b_j$ is the activation of the $j$th neuron. $y_j$ is used to symbolize the softmax output of the $j$th neuron, as illustrated in Figure 1 (c), which is also considered as the output after the soft WTA operation. It can be shown that the synaptic weight vector $w_j$ will be implicitly normalized by the learning rule to a unit vector when $\langle x, w_j \rangle \geq 0$. To make this point clear, let's consider the derivative of the norm of $w_j$

$$\frac{d\|w_j\|^2}{dt} = 2w_j \frac{dw_j}{dt} = 2\frac{\eta}{\tau} u_j y_j \cdot (1 - \|w_j\|^2). \tag{6}$$

Provided that $u_j y_j \geq 0$, the derivative of the vector norm increases if $\|w_j\|^2 < 1$, and decreases otherwise if $\|w_j\|^2 > 1$. Since $y_j \geq 0$, the weight vector tends to converge to a unit vector when $u_j \geq 0$. The learning rule in equation 3 is similar to the *Oja* rule. The difference is *Oja* rule only considers linear weighted summation of inputs $y_j$ for neuron $j$; equation 3 considers both linear weighted summation of inputs $u_j$ and nonlinear output $y_j$ for neuron $j$. While the activation on neuron $j$ is large enough that $y_j$ is close to 1, equation 3 reduces to the learning rule in *BioHash*.

In the following content, we will analyze the differences between *BioHash* and *SoftHash* in view of machine learning, to explain that learning with the soft WTA mechanism is more effective in

capturing the weak correlations among data samples. First, the learning process can be shown to minimize the following energy function, defined by

$$E = - \sum_{x \in X} \sum_{j=1}^{m} y_j f \left( \frac{<x, w_j>}{<w_j, w_j>^{\frac{1}{2}}} \right). \tag{7}$$

It is a monotonically decreasing function (More details in **Appendix** A), $f(\cdot)$ is an exponential function. Note that the synaptic plasticity rule in equation 3 is local; the learning process does not perform gradient descent, i.e. $\dot{w}_j \neq \nabla_{w_j} E$.

In (Ryali et al., 2020), the energy function of *BioHash* with the specific settings is approximate to the spherical *K*-means clustering algorithm (More details in **Appendix** B), where the distance from the input point to the centroid is measured by the normalized inner product instead of the Euclidean distance. Thus, each input sample can belong to only one neuron, as illustrated in Figure 2 (a).

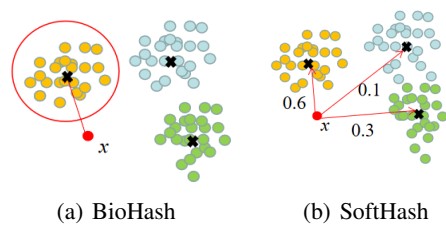

(a) BioHash (b) SoftHash

Figure 2: Illustration of approximate clustering ways of learning processes of *BioHash* and *SoftHash*.

Obviously, the energy function equation 7 can be approximated as a mixture probabilistic model, where each output neuron is treated as a separate probabilistic model. In this framework, the clustering process can be interpreted as the probability that each input sample belongs to different neurons Figure 2 (b). Consequently, each input sample can be associated with one or multiple neurons, allowing for a more flexible and nuanced representation of the data. This probabilistic modeling approach in *SoftHash* enables a richer understanding of semantic information from input data, and captures complex patterns and associations in the data, leading to more expressive and discriminative hash codes in the next subsection.

## 3.2 HASHING ALGORITHM

After the learning process is complete, only the highest firing neurons retain their values as one, and the others are zeroed out. The hash code is generated in the following way, which sparsifies the representation while preserving the largest and most discriminative coefficients. For a given sample $x$, we generate a hash code $v \in \{0, 1\}^m$ as

$$v_j = \begin{cases} 1, & y_j \text{ is in top } k \\ 0, & otherwise. \end{cases} \tag{8}$$

$v$ is a high-dimensional sparse binary hash code ( a vector of *m* elements, with *k* ones in it), where *k* is denoted as hash length.

**Biological Interpretation** The process of generating hash codes via *SoftHash* is a crude mathematical approximation of the computation performed by the PN-KC-APL neural networks in Figure 1 (a). An input sample $x$ generates its posterior probabilities into the KC neurons using feedforward synaptic weights $W$ and neuron biases $b$. The KC-APL recurrent network retains a small fraction of activated KCs and silences the remaining. Finally, the activated KCs are assigned state 1 while the remaining inactive KCs are assigned state 0. The overall algorithm of *SoftHash* is shown in **Appendix** C.

We provide intuition behind *SoftHash* to explain the locality-sensitive property in it. For the purpose of fast similarity search, the location of an input data sample $x$ can be specified by observing the nearest few references' from a set of *m* reference points that are picked in the hash space, producing a sparse and useful local representation. Such that, the nearest neighbors of the input data be quickly found by the hamming distance. We model all output neurons as probabilistic models to support the entirety of the data distribution. The values on the output layer represent the probabilities that each input sample belongs to them, which are calculated based on input data, synaptic weights, and neuron biases. The plasticity rule for synaptic weights in equation 7 and the iterative update

for neuron biases in equation 4 assign $m$ probabilistic models, enabling their density to match the input data density. In other words, more output neurons are needed to provide high resolution where the data density is high; fewer neurons are needed where the data density is low. In this way, our algorithm has the sufficient capability to enable similar inputs to activate similar output neurons and be clustered closely in a new dimension extended hash space.

### 3.3 COMPUTATIONAL COMPLEXITY ANALYSIS

The two most time-consuming parts of our proposed *SoftHash* are the update steps of synaptic weights and neuron biases. For each epoch, the time complexity of performing the dot product on synaptic weights is $N \cdot d \cdot m$, and that of update neuron biases is $N \cdot m$. The softmax formula costs $N \cdot (m+1) \cdot m$ operations, and equation 3 requires $d \cdot m$ in addition to $N \cdot d \cdot m$ operations for calculating the dot-product for each input data sample. Thus, the overall computational complexity of our approach is $O(N \cdot d \cdot m + N \cdot m + N \cdot (m+1) \cdot m + d \cdot m) \approx N \cdot d \cdot m + N \cdot m^2$ per epoch. For the storage cost, the sparse binary representation obtained by *SoftHash* entails the storage of $k log_2 m$ bits per data sample and $O(k)$ computational cost to compute Hamming distance. Empirically, in the next sections, we will show *SoftHash* can preserve the data correlation very well for image and text datasets.

## 4 IMAGE RETRIEVAL

In this section, we use the image retrieval task to evaluate the performance of *SoftHash*. The image retrieval aims to find the most similar ones of any given image from a large image database, which is an important task in machine learning and information retrieval.

### 4.1 EXPERIMENT SETTINGS

**Datasets.** We conduct experiments on three public benchmark datasets for image retrieval, including **Fashion-MNIST** (Xiao et al., 2017), **CIFAR10** (Krizhevsky et al., 2009) and **CIFAR100**. In specific, **Fashion-MNIST** contains 70k grey-scale images (size $28 \times 28$) of clothing products from 10 classes. **CIFAR10/CIFAR100** contain 60k RGB images (size $32 \times 32 \times 3$) of animals and vehicles from 10/100 classes. For datasets **Fashion-MNIST** and **CIFAR10**, we randomly select 100 query inputs of each class to form a query set of 1,000 images (Dasgupta et al., 2017). For **CIFAR100**, we randomly select 10 query inputs of each class to obtain 1,000 query images. The remaining images are used for both training as well as the database for retrieval.

**Evaluation Metrics.** Following previous work Su et al. (2018), we adopt the widely used **Mean Average Precision (MAP)** to measure the hamming ranking quality.

**Baseline Methods.** We compare our proposed *SoftHash* with data-dependent hashing algorithms Spectral Hashing (*SH*) (Weiss et al., 2008), Iterative Quantization (*ITQ*) (Gong et al., 2012) and Scalable Graph Hashing (*SGH*) (Jiang & Li, 2015), the hashing algorithm with binary random weights *FlyHash* (Dasgupta et al., 2017), and the hashing algorithm with learnable synaptic weights *Bio-Hash* (Ryali et al., 2020). Besides, the conventional hashing algorithm *LSH* is also included in our study. It generates the hash code of the input data sample $x$ according to $y = sgn(Mx)$, where $M \in \mathbb{R}^{m \times k}$ is a random matrix. $k$ is the hash length.

### 4.2 RESULTS AND DISCUSSION

**Comparison results** The mAP@1000 results of different hashing algorithms on three benchmarks are shown in Table 2, with hashing lengths varying from 32 to 128. From the table, it is evident that *SoftHash*-2 demonstrates the best retrieval performance among all the hashing algorithms, particularly at larger hash lengths, especially when $k = 128$. The performance of *SoftHash*-2 significantly improves with the increase in hash length compared to *SoftHash*-1. Comparing the results of *Fly-Hash* and *LSH*, it is clear that *FlyHash* outperforms the classical algorithm *LSH* at different hash lengths, indicating the superiority of high-dimensional sparse binary representations.

**Ablation study** We conduct experiments to evaluate the mAP@1000 results of *BioHash* and *Soft-Hash* with different weight initialization settings. We can observe that both *SoftHash*-2 and *Bio-*

Table 2: mAP@1000 (%) on **Fashion-MNIST**, **CIFAR10** and **CIFAR100**, ground truth based on *Euclidean distance*, following protocol in (Li et al., 2018) (Ryali et al., 2020). The best results for each hash length are shown in **boldface**. *SoftHash*-1 (*BioHash*-1) and *SoftHash*-2 (*BioHash*-2) have different weight initialization settings, uniform initialization for *SoftHash*-1 (*BioHash*-1) and Gaussian initialization for *SoftHash*-2 (*BioHash*-2). More details are summarized in Appendix D.2.

| Method | Fashion-MNIST | | | CIFAR10 | | | CIFAR100 | | |
|---|---|---|---|---|---|---|---|---|---|
| | 32 | 64 | 128 | 32 | 64 | 128 | 32 | 64 | 128 |
| *LSH* | 0.1734 | 0.2506 | 0.3348 | 0.0710 | 0.1054 | 0.1446 | 0.0683 | 0.0998 | 0.1375 |
| *SH* | 0.3464 | 0.4223 | 0.4675 | 0.1310 | 0.1471 | 0.1819 | 0.1096 | 0.1557 | 0.1870 |
| *ITQ* | 0.3133 | 0.4159 | 0.4913 | 0.1763 | 0.2209 | 0.2605 | 0.1672 | 0.2128 | 0.2552 |
| *SGH* | 0.3095 | 0.3488 | 0.3941 | 0.1133 | 0.1362 | 0.1690 | 0.1245 | 0.1560 | 0.1869 |
| *FlyHash* | 0.3623 | 0.4203 | 0.4595 | 0.1470 | 0.1790 | 0.2037 | 0.1399 | 0.1684 | 0.1886 |
| *BioHash*-1 | 0.3876 | 0.4031 | 0.4101 | 0.1974 | 0.2115 | 0.2235 | 0.2024 | 0.2109 | 0.2235 |
| *BioHash*-2 | 0.4025 | 0.4411 | 0.4586 | 0.2054 | 0.2716 | 0.3211 | 0.2254 | 0.2863 | 0.3328 |
| *SoftHash*-1 | 0.4750 | 0.5530 | 0.6028 | 0.2164 | 0.2685 | 0.3024 | 0.2266 | 0.2710 | 0.3084 |
| *SoftHash*-2 | **0.5080** | **0.6003** | **0.6712** | **0.2632** | **0.3570** | **0.4681** | **0.2872** | **0.4038** | **0.5163** |

*Hash*-2, when initialized with synaptic weights following the standard normal distribution, achieve better results. The reason may be that Gaussian initialization is more effective in breaking symmetry compared to uniform initialization. In this way, Gaussian initialization ensures that each neuron learns distinct features from the beginning, and improves the network's ability to capture input data characteristics during the learning process.

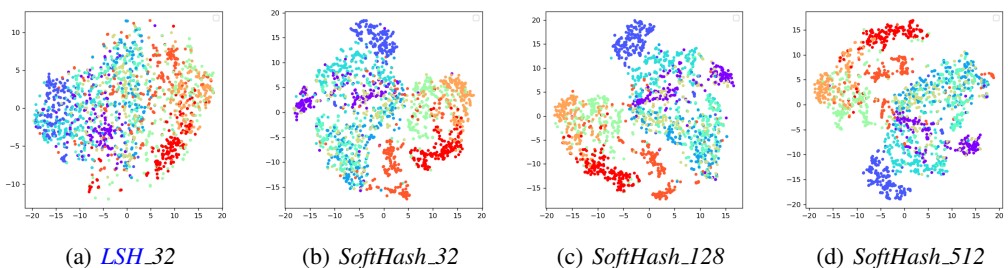

(a) *LSH_32*  (b) *SoftHash_32*  (c) *SoftHash_128*  (d) *SoftHash_512*

Figure 3: tSNE embedding of *LSH* (Hash Length $k = 32$) and *SoftHash-2* (Hash Length $k = 32$, 128, 512) on the dataset **Fashion-MNIST**.

**Visualization** We randomly selected 2,000 input data samples from the **Fashion-MNIST** dataset and visualized the geometry of the hash codes obtained by *LSH* and *SoftHash-2* using t-Stochastic Neighbor Embedding (t-SNE) (Van der Maaten & Hinton, 2008). The visualization is shown in Figure 3, where each point represents an input sample, and the color indicates the category it belongs to. From subfigures (a) and (b) in Figure 3, it is obvious that the cluster structure obtained by *SoftHash-2* is more compact compared to *LSH* when using the same hash length. This suggests that *SoftHash-2* generates hash codes that result in tighter and more distinct clusters in the embedding space. Furthermore, as depicted in subfigures (b), (c), and (d) of Figure 3, increasing the hash length of *SoftHash-2* refines the manifold structure. In other words, input data samples belonging to the same category exhibit smaller distances from each other in the hash code space. These visualizations provide evidence of the effectiveness of *SoftHash-2* in generating compact and discriminative hash codes that capture meaningful information in the data.

## 5 WORD SIMILARITY SEARCH

Word embeddings contain the semantic and syntactic information of words, which are represented with dense vectors. Existing post-process operations Yogatama & Smith (2015); Wang et al. (2019b) have tried to transform original word embeddings into sparse ones, speeding up computation. In this section, we take word embeddings provided by the pre-train language model as input data and apply

hashing algorithms to get their binary representations, we will conduct experiments to evaluate the performance of binary embeddings for word similarity search.

## 5.1 Experiment Settings

**Datasets.** We use the pre-trained **GloVe** embeddings Pennington et al. (2014) as original input data. We conduct experiments on three public word similarity datasets: **WordSim353** Agirre et al. (2009), **SimLex999** Hill et al. (2015), **Men** Bruni et al. (2014), which contain 353, 999, 3,000 pairs of words with human-annotated similarity scores, respectively.

**Evaluation Metrics.** We use cosine similarity Liang et al. (2021) to evaluate similarity scores for original real-valued representations from **GloVe** embeddings ($d$= 300). Following previous work Tissier et al. (2019), the similarity scores for binary representations are evaluated by the Hamming similarity.

## 5.2 Results and Discussion

The Spearman's rank correlation scores obtained with the similarity and human-annotated scores are depicted in Figure 4. Each sub-figure corresponds to different datasets, and the horizontal axis represents the hash lengths. For *SoftHash*, the best scores are achieved at hash lengths of k=128 or k=512, which are very close to the scores obtained with the original dense vectors provided by **GloVe**. For *LSH*, the results increase with hash length varying from 2 to 128, but the deviation is large compared to **GloVe**. For *BioHash*, the results increase rapidly with the increase of the hash length when the hash length is small, but the results no longer improve when hash lengths reach certain values, which are *k*=16 for **SimLex999**, *k*=64 for **WordSim353**, and *k*=64 for **Men**. From Figure 4, we could see that sparse binary representations obtained by *SoftHash* achieve similar results to original vectors from **GloVe** on the word similarity task.

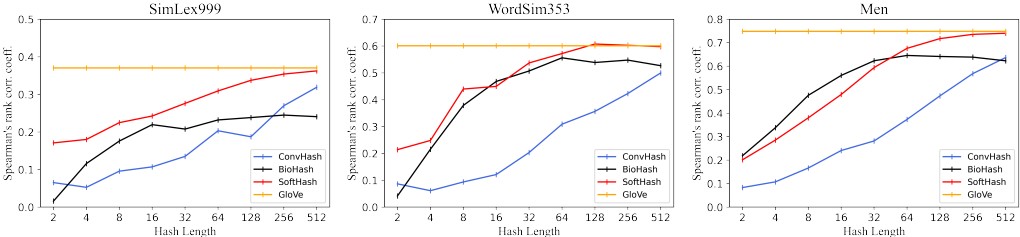

Figure 4: Performance comparison of hashing algorithms with different hash lengths on the word similarity task via Spearman's rank correlation coefficient.

In Figure 5, we show the computation time on the original embedding from **GloVe** and binary codes obtained from different hashing algorithms. It is observed that searching with binary representations is faster than searching with the original dense vectors, further highlighting the computational advantages of using binary embeddings. Overall, these findings suggest that *SoftHash* outperforms *LSH* and *BioHash* in preserving relative correlation, while providing computational efficiency through binary representations.

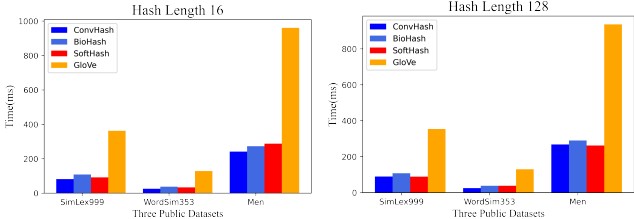

Figure 5: Comparison of computation time of different output representations on **SimLex999**, **WordSim353** and **Men** with a single CPU core.

We conduct the experiment that finds the nearest neighbor words of a target word for further observing whether hash codes can catch the semantic information of word embeddings. The ground truth

in this experiment is the top 10 nearest neighbor words of a target word, which are measured by the cosine similarity between embeddings of words. Figure 6 presents the top 10 nearest neighbor words for the target words 'apple' and 'women' in the hash code space, with hash lengths of 64 and 512, respectively. The results demonstrate that *SoftHash* successfully preserves the semantic information similar to the original word embeddings. Notably, in Figure 6, we could observe that hash codes generated by *SoftHash* not only identify similar nearest neighbor words for the target word 'women' but also maintain the same relative order of the words. We define the $Rel\_score$ ($Rel\_score \in [0,1]$) to observe the performance of the top 10 nearest neighbor words of a target word of different hashing algorithms (more detail in **Appendix** E.2). The higher $Rel\_score$ represents results more similar to the ground truth provided by **GloVe**. This observation further supports the effectiveness of *SoftHash* in capturing and preserving semantic relationships between words.

| word | method | 10 nearest neighbor words | Rel_score |
|------|--------|---------------------------|-----------|
| apple | GloVe | iphone, macintosh, ipod, microsoft, ipad, intel, ibm, google, software, motorola | 1.0 |
| | ConvHash | crop, regardless, outset, whichever, apples, farmer, fruit, nonpartisan, cider, landing | - |
| | BioHash | chips, banana, potato, coffee, beer, strawberry, apples, chip, cheese, fruit | - |
| | SoftHash | **ibm**, **microsoft**, adobe, **macintosh**, **ipod**, **ipad**, mango, **intel**, **iphone**, hardware | 0.738 |
| women | GloVe | men, girls, female, woman, male, mothers, athletes, she, young, children | 1.0 |
| | ConvHash | **woman**, teenagers, look, cry, remember, **children**, elderly, treatment, deserve, all | 0.246 |
| | BioHash | **men**, ladies, **girls**, **woman**, **athletes**, **female**, children, compete, **mothers**, families | 0.782 |
| | SoftHash | **men**, **girls**, **female**, youth, **children**, **woman**, wives, **athletes**, **male**, aged | 0.855 |

(a) Hash length 64

| word | method | 10 nearest neighbor words | Rel_score |
|------|--------|---------------------------|-----------|
| apple | GloVe | iphone, macintosh, ipod, microsoft, ipad, intel, ibm, google, software, motorola | 1.0 |
| | ConvHash | **macintosh**, **iphone**, **google**, **ipod**, fruit, mac, blackberry, netscape, strawberry, citrus | 0.676 |
| | BioHash | chip, chips, sells, sold, buy, **iphone**, **ipod**, cherry, store, maker | 0.208 |
| | SoftHash | **iphone**, **macintosh**, **ipod**, **ibm**, **microsoft**, **intel**, **ipad**, cider, **motorola**, **google** | 0.970 |
| women | GloVe | men, girls, female, woman, male, mothers, athletes, she, young, children | 1.0 |
| | ConvHash | **men**, **female**, **male**, **girls**, **woman**, **mothers**, **she**, ladies, **young**, girl | 0.935 |
| | BioHash | **men**, **girls**, **female**, **young**, **woman**, ladies, **athletes**, who, **children**, youth | 0.804 |
| | SoftHash | **men**, **girls**, **female**, **woman**, **male**, **athletes**, ladies, **young**, **children**, **she** | 0.952 |

(b) Hash length 512

Figure 6: For each target word (left), 10 nearest neighbor words (middle) and $Rel\_score$ (right) of hashing algorithms with different hash lengths are shown. The nearest neighbor words provided by **GloVe** are ground truth. The nearest neighbor words provided by hashing algorithms that appear in the ground truth are shown in **boldface**.

## 6 CONCLUSIONS

In this work, we propose an unsupervised hashing algorithm *SoftHash* to generate discriminative and sparse hash codes. *SoftHash* is a data-dependent algorithm, its learnable synaptic weights and neuron biases are updated in a neurobiologically plausible way. In particular, we adopt a soft WTA mechanism in the learning process, allowing weakly correlated data a chance to be learned to efficiently generate more representative high-dimensional hash codes. We demonstrate that *SoftHash* outperforms recent data-driven hashing algorithms for image retrieval on benchmark datasets. Additionally, we explore the application of *SoftHash* in post-processing word embeddings for word similarity search. Our experiments show that our sparse binary representations maintain the rank correlation of the original dense embeddings while requiring less computation time.

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

# A   MORE DETAILS ABOUT THE ENERGY FUNCTION

We show the time derivative of the energy function equation 7 for the special case that only one neuron $j$ active ($y_j \approx 1$)

$$
\tau_E \frac{dE}{dt} = - \sum_{x \in X} \frac{f(\langle x, \frac{w_j}{\|w_j\|} \rangle)}{<w_j, w_j>^{\frac{3}{2}}} \left[ \tau_E \langle \frac{dw_j}{dt}, x \rangle \langle w_j, w_j \rangle - \tau_E \langle w_j, x \rangle \langle \frac{dw_j}{dt}, w_j \rangle \right] =
$$
$$
- \sum_{x \in X} \tau_E \frac{f(\langle x, \frac{w_j}{\|w_j\|} \rangle)}{<w_j, w_j>^{\frac{3}{2}}} \left[ \langle x, x \rangle \langle w_j, w_j \rangle - \langle w_j, x \rangle^2 \right] \leq 0, \tag{9}
$$

where $f(\cdot)$ is an exponential function, so $f(\langle x, \frac{w_j}{\|w_j\|} \rangle) > 0$, $\tau_E$ is the time constant. The final expression is obtained according to the Cauchy-Schwarz inequality.

# B   *BioHash* VS. *SoftHash*

## B.1   CONNECTIONS AND DIFFERENCES

*BioHash* (Ryali et al., 2020) and *SoftHash* are all data-dependent bio-inspired hashing methods. Both of them learn the hashing projection function using the Hebbian-like learning rule coupled with the idea of winner-take-all.

*BioHash* adopts the Hebbian rule combined with a hard WTA mechanism to learn synaptic weights from data. The learning of *BioHash* can be formalized as minimizing the following energy function

$$
E = - \sum_{x \in X} \sum_{j=1}^{m} g[Rank(<w_j, x>_j)] \frac{<w_j, x>_j}{<w_j, w_j>_j^{\frac{p-1}{p}}} \tag{10}
$$

where $<X, Y>_j = \sum_{i,j} g_{ij}^j X_i Y_j$, with $g_{ij}^j = |w_i|^{p-2} \delta_{ij}$, $p \geq 1$ is a Lebesgue norm hyper-parameter (Krotov & Hopfield, 2019) and $\delta_{ij}$ is Kronecker delta. The *Rank* operation sorts the inner products from the largest ($j = 1$) to the smallest ($j = m$), and

$$
g[j] = \begin{cases} 1, & j = 1 \\ -\Delta, & j = 2 \\ 0, & otherwise. \end{cases} \tag{11}
$$

For $p$=2 and $\Delta$=0 (Ryali et al., 2020), the energy function of *BioHash* is approximate to the spherical $K$-means clustering algorithm, where the distance from the input point to the centroid is measured by the normalized inner product instead of the Euclidean distance.

In equation 10, it is clear that each input sample belongs to only one class. It means only the output neuron with the largest value can be updated during the learning process, limiting its ability to catch the semantic information of the input data. *SoftHash* alleviates this limitation by introducing a soft WTA mechanism for learning the hash function. In contrast to *BioHash* using a hard WTA mechanism, our method makes non-winning classes not fully suppressed in the learning process. By assigning a credit score $\frac{e^{u_j+b_j}}{\sum_{l=1}^{m} e^{u_l+b_l}}$ in Eq (5) to each class, *SoftHash* enables the weakly correlated classes to have a chance to influence the representative learning, making the learned hash codes improved with more semantic information.

## B.2   EXPERIMENT AND DISCUSSION

We make a further comparison between *BioHash* and *SoftHash* on the image retrieval task, where the hash length varies from 2 to 512. The mAP@1000 results of *BioHash* and *SoftHash* on **Fashion-MNIST**, **CIFAR10** and **CIFAR100** are shown in Figure 7.

From the figure, it is evident that *SoftHash*-2 (red line) demonstrates the best retrieval performance among all the hashing algorithms, especially when $k = \{64, 128, 256, 512\}$. Regarding *BioHash*-1 (dotted blue line), they exhibit the best retrieval performance at small hash lengths. However, there is only a slight improvement from $k = 32$ to $k = 64$ and an even smaller improvement from $k = 64$ to $k = 512$. These results align with the conclusions presented in the corresponding work Ryali et al. (2020).

In summary, we suggest the following:

- *SoftHash*: *SoftHash* is recommended when the primary objective is to maintain the relative relationships between the original data points as accurately as possible while projecting them into hash space. It is a particularly good choice for scenarios where semantic information and similarity structure preservation are essential.

- *BioHash*: *BioHash* is a suitable choice in situations where there is a requirement for short hash lengths. It excels in achieving excellent performance at small hash lengths which can be advantageous for scenarios where storage is a limiting factor.

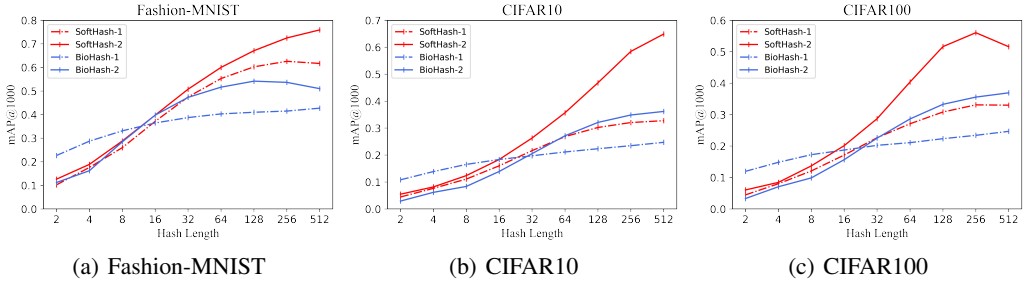

(a) Fashion-MNIST        (b) CIFAR10        (c) CIFAR100

Figure 7: Illustration of the mAP@1000 results of *SoftHash* and *BioHash* on three datasets.

## C ALGORITHM

The pseudo-codes of generating hash codes are illustrated in Algorithm 1.

---

**Algorithm 1** SoftHash

---

**Require:** synaptic weights $W$, biases $b$, hash length $k$

    **for** $x \in X$ **do**

        **for** $j \in \{1, 2, ..., m\}$ **do**

            $u_j = \langle x, w_j \rangle = \sum_i w_j^i x_i$

        **end for**

        **for** $j \in \{1, 2, ..., m\}$ **do**

            $y_j = softmax(u_j + b_j)$

        **end for**

        **for** $j \in \{1, 2, ..., m\}$ **do**

            **if** $y_j$ in top $k$ of all activations **then**

                $v_j = 1$

            **else**

                $v_j = 0$

            **end if**

        **end for**

        **return** $v$

    **end for**

---

# D   MORE DETAILS ABOUT IMAGE RETRIEVAL

## D.1   IMPLEMENTATION DETAILS

Following the protocol in Dasgupta et al. (2017), for datasets **Fashion-MNIST** and **CIFAR10**, we randomly select 100 query inputs of each class to form a query set of 1,000 images. For CIFAR100, we randomly select 10 query inputs of each class. The remaining images are used for both training as well as the database for retrieval. Each input image is normalized to be a unit vector in the 784 (28×28) dimensional space and the 3072 (32×32×3) dimensional space for **Fashion-MNIST** and **CIFAR10/CIFAR100**, respectively. Ground truth is the top 1000 nearest neighbors of a query in the database, based on Euclidean distance between pairs of images in pixel space.

We set the **output dimensions to 2,000** for high-dimensional representations. For *ConvHash* and *FlyHash*, we average the mAP over 10 trials, where the random matrix changes in each trial. We set the sampling ratio of input data samples to 0.1 for *FlyHash*, following the previous workDasgupta et al. (2017). For hashing algorithms with learnable synaptic weights, we set the initial learning rate to 0.4 for **Fashion-MNIST**, which decays from 0.04 to 0 during the learning process; and we set the learning rate to 0.2 for **CIFAR10/CIFAR100**, which also decays from 0.02 to 0 as in the case of **Fashion-MNIST**. For *BioHash*, the training is done for 100 epochs with mini-batches of size 100 on three datasets. The Lebesgue norm is set to $p$=4 and the anti-Hebbian learning parameters $\Delta$ are set to 0.4 and 0.3 for **Fashion-MNIST** and **CIFAR10/CIFAR100**Krotov & Hopfield (2019), respectively. For *SoftHash*, the training is done for 20 epochs, the batch sizes are all 1024 on three datasets. Biases are initialized following a negative uniform distribution. The weight initialization methods for *BioHash* and *SoftHash* are shown in **Appendix** D.2. Additionally, we adopt temperature scaling Hinton et al. (2015), a mechanism that can maintain the probabilistic interpretation of the softmax output, to scale exponent by $y_\mu = \frac{e^{(u_\mu + b_\mu)/T}}{\sum_{l=1}^{m} e^{(u_l + b_l)/T}}$, $T$ is set to 10.

The initialization methods for *BioHash* and *SoftHash* are summarized in Table 3.

Table 3: Initialization settings for *BioHash* and *SoftHash*.

|  | **Weight initialization** | **Bias initialization** |
|---|---|---|
| *BioHash*-1 | uniform distribution | - |
| *BioHash*-2 | standard normal distribution | - |
| *SoftHash*-1 | uniform distribution | uniform distribution |
| *SoftHash*-2 | standard normal distribution | uniform distribution |

## D.2   MORE EXPERIMENTAL RESULTS

We evaluate the performance of hashing algorithms for image retrieval, with hash lengths varying from 2 to 512. Experimental results on three benchmarks are shown in Figure 8. It is evident that *SoftHash*-2 consistently demonstrates the best retrieval performance among all the hashing algorithms, particularly at large hash lengths.

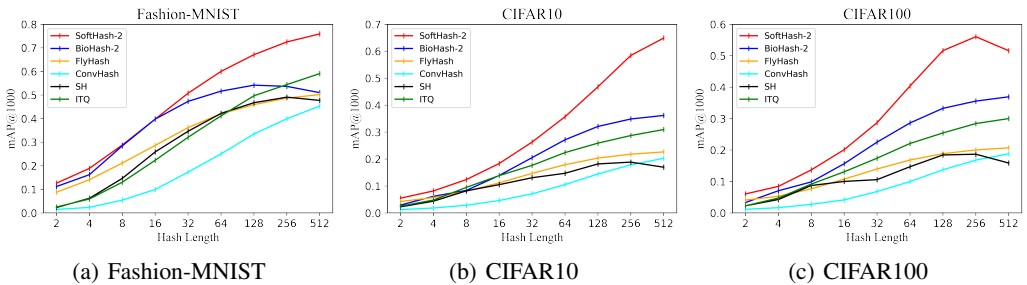

(a) Fashion-MNIST          (b) CIFAR10          (c) CIFAR100

Figure 8:   Illustration of the mAP@1000 results of different hashing algorithms on **Fashion-MNIST**, **CIFAR10** and **CIFAR100**.

# E MORE DETAILS ABOUT WORD SIMILARITY SEARCH

## E.1 IMPLEMENTATION DETAILS

We select the top 50,000 most frequent words from pre-trained **GloVe** embeddings and use them to train synaptic weights for *BioHash* and *SoftHash*. We set the **output dimensions to 2,000** for high-dimensional representations. We set the initial learning rate to 0.4, which decays from 0.04 to 0 during the learning process. For *ConvHash*, we average the mAP over 5 trials, where the random matrix also changes in each trial. For *BioHash*, the Lebesgue norm is set to $p$=4 and the anti-Hebbian learning parameter $\Delta$ is set to 0.4. Weights are initialized following a standard normal distribution for *BioHash* and *SoftHash*. The remaining settings are the same as those used for image retrieval.

## E.2 DETAILS OF FINDING NEAREST WORDS

The Normalized Discounted Cumulative Gain ($NDCG$) is a widely used measure of ranking quality Wei et al. (2018). NDCG takes into account the position of relevant items in a ranked list, emphasizing that items higher in the ranking should receive more credit than those lower in the ranking. Following NDCG, we define the $Rel\_score$ to evaluate the performance of the top 10 nearest neighbor words of a target word of different hashing algorithms, which is calculated by

$$Rel\_score = \frac{Rel\_value(*)}{Rel\_value(GloVe)} \tag{12}$$

where $Rel\_value(*)$ denotes the $Rel\_value$ of the hashing algorithm, formulating by

$$Rel\_value(*) = \sum_{Index=1}^{10} \frac{Rel(Index)}{log_2(Index+1)}. \tag{13}$$

In this equation, the contribution of each word to $Rel\_value(*)$ is measured based on two factors: the relevance score and its position($Index$) in the ranked list. $Rel(Index)$ denotes the relevance score of the word positioned at $Index$, which is determined by the results of **GloVe**. In this study, the relevance scores of the top 10 nearest neighbor words provided by **GloVe** are set to $\{Rel(1),$ $Rel(2), ..., Rel(10) = 1.0, 0.9, ..., 0.1\}$. Additionally, $log_2(Index+1)$ is used to discount the relevance score of each word. Note that words not found in the ground truth have a relevance score of 0. In Figure 9, we provide an example of how to calculate the $Rel\_score$ of *SoftHash* for the target word "apple".

| Index | 1 | 2 | 3 | 4 | 5 | 6 | 7 | 8 | 9 | 10 |
|---|---|---|---|---|---|---|---|---|---|---|
| GloVe | iphone | macintosh | ipod | microsoft | ipad | intel | ibm | google | software | motorola |
| *Rel*(Index) | 1.0 | 0.9 | 0.8 | 0.7 | 0.6 | 0.5 | 0.4 | 0.3 | 0.2 | 0.1 |
| SoftHash | **iphone** | **machintosh** | **ipod** | **ibm** | **microsoft** | **intel** | **ipad** | cider | **motorola** | **google** |
| *Rel*(Index) | 1.0 | 0.9 | 0.8 | 0.4 | 0.7 | 0.5 | 0.6 | 0 | 0.1 | 0.3 |
| $log_2$ (Index + 1) | 1.00 | 1.59 | 2.00 | 2.32 | 2.59 | 2.81 | 3.00 | 3.17 | 3.32 | 3.46 |

$$Rel\_value(\text{GloVe}) = \frac{1.0}{1.00} + \frac{0.9}{1.59} + \frac{0.8}{2.00} + \frac{0.7}{2.32} + \frac{0.6}{2.59} + \frac{0.5}{2.81} + \frac{0.4}{3.00} + \frac{0.3}{3.17} + \frac{0.2}{3.32} + \frac{0.1}{3.46} \approx 2.99$$

$$Rel\_value(\text{SoftHash}) = \frac{1.0}{1.00} + \frac{0.9}{1.59} + \frac{0.8}{2.00} + \frac{0.4}{2.32} + \frac{0.7}{2.59} + \frac{0.5}{2.81} + \frac{0.6}{3.00} + \frac{0}{3.17} + \frac{0.1}{3.32} + \frac{0.3}{3.46} \approx 2.90$$

$$Rel\_score = Rel\_value(\text{SoftHash})/Rel\_value(\text{GloVe}) = \frac{2.90}{2.99} \approx 0.970$$

Figure 9: An example of calculating $Rel\_score$ of *SoftHash* for the target word "apple".

