# OpenReview forum: "SoftHash: High-dimensional Hashing with A Soft Winner-Take-All Mechanism"
_ICLR.cc/2024/Conference — Submitted to ICLR 2024_

### Official Review · Reviewer_3Wy1 · 2023-10-15

**Soundness:** 3 good
**Presentation:** 3 good
**Contribution:** 3 good
**Rating:** 5
**Confidence:** 5

**Summary:**

Inspired by the bio-neverous system of fruit fly and existing work bio-hash, this paper proposed a data-dependent hashing method dubbed SoftHash, which is characterized in three aspects: (1) a novel algorithm what can generate sparse and yet discriminative high-dimensional hash codes; (2) a mechanism that combines a Hebbian-like local learning rule and the soft WTA; (3) evaluation on image retrieval and word similarity search tasks.

**Strengths:**

(1) A novel algorithm: The proposed SoftHash is interesting with simple and effective interpretable learning mechanism.
(2) Solid experiments: two different kinds of tasks, image retireval and word similarity search, are designed to validate SoftHash's good performance in compact semantic embeddings.

**Weaknesses:**

(1) There are many places that were not clearly explained, such as, what is ConvHash? what is the difference between SoftHash-1 and Soft-Hash-2?
(2) Why not just choose SH and ITQ? there are more similar papers such as SGH [1], DGH [2], COSDISH [3]?
(3) W.r.t. others, please refer to the Question part.


[1] Qing-Yuan Jiang, Wu-Jun Li: Scalable Graph Hashing with Feature Transformation. IJCAI 2015: 2248-2254
[2] Wei Liu, Cun Mu, Sanjiv Kumar, Shih-Fu Chang: Discrete Graph Hashing. NIPS 2014: 3419-3427
[3] Wang-Cheng Kang, Wu-Jun Li, Zhi-Hua Zhou: Column Sampling Based Discrete Supervised Hashing. AAAI 2016: 1230-1236

**Questions:**

I have several questions listed as follows:
(1) What is the main differences between the bio-inspired hashing methods with sparse-and-high-dimensional {0,1}-vectors and the conventional hashing methods with dense-and-low-dimensional {0,1}-vectors?
(2) How does the bio-inspired hashing codes capture the data samples' semantics?
(3) With respect to the storage and computation, how does the bio-inspired hashing approaches realize economic memory and fast retrieval/semantic computing?

---

> ### Author Response · Authors · 2023-11-20
>
> Thank you for your time and helpful comments. We address the questions and concerns below:
>
> R-W1: Thanks for your detailed comments. In this paper, ConvHash is the classical LSH method, it generates the hash code of the input data sample $x$ according to $y = sgn(Mx)$, where $\textit{M} \in \mathbb{R}^{m \times k}$ is a random matrix. $k$ is the hash length. We have changed 'ConvHash' to 'LSH' for simplicity. SoftHash-1 and SoftHash-2 have different weight initialization methods, uniform initialization for SoftHash-1, and Gaussian initialization for SoftHash-2. We have added a description of them in Table 2 of the paper.
>
> R-W2: Thanks for your comments and recommendation. In this paper, we extend our comparison beyond conventional methods like SH and ITQ, as our proposed SoftHash is an unsupervised bio-inspired hashing method. Therefore, our experiment includes a comparison with other bio-inspired hashing methods. SGH, DGH and COSDISH are well-known hashing methods. We are conducting experiments to evaluate the performance of SGH on image retrieval. Other works have been concluded in the introduction and related works of the paper. Currently, experimental results are shown in the Table R-W2.
>
> Table R-W2
>
> |        |        | Fashion-MNIST |        |        | CIFAR10 |        |        | CIFAR100 |        |
> | :----: | :----: | :-----------: | :----: | :----: | :-----: | :----: | :----: | :------: | :----: |
> | Method | 32     | 64            | 128    | 32     | 64      | 128    | 32     | 64       | 128    |
> | LSH    | 0.1734 | 0.2506   | 0.3348 | 0.0710 | 0.1054  | 0.1446 | 0.0683 | 0.0998   | 0.1375 |
> | SH     | 0.3464 | 0.4223   | 0.4675 | 0.1310 | 0.1471  | 0.1819 | 0.1096 | 0.1557   | 0.1870 |
> | ITQ    | 0.3133 | 0.4159   | 0.4913 | 0.1763 | 0.2209  | 0.2605 | 0.1672 | 0.2128   | 0.2552 |
> | SGH    | 0.3095 | 0.3488   | 0.3941 | 0.1133 | 0.1362  | 0.1690 | 0.1245 | 0.1560   | 0.1869 |
>
>
> R-Q1: The main differences are
> (1) the bio-inspired hashing methods project input data into high-dimensional representations, whereas conventional hashing methods map the data into low-dimensional representations;
> (2) the bio-inspired hashing methods sparsify the high-dimensional representation using a winner-take-all mechanism, whereas conventional hashing methods preserve a dense representation.
> In general, the high-dimensional sparse representations are always more robust to encoding noisy input data [1].
> With the help of the bio-winner-take-all mechanism, the representations can also achieve a good performance in semantic clustering.
>
> [1] How can we be so dense? The Benefits of Using Highly Sparse Representations
>
> R-Q2: Most of the bio-inspired hashing codes are generated with the winner-take-all mechanism.
> The kernel assumption of the winner-take-all mechanism is that with an input vector $x$, if the semantic information of attribute $x_i$ is dominant over some other attribute, $x_i$ has stronger discriminative powers to represent $x$ and cluster $x$ [1]. This allows the most semantical features can be more utilized than other noisy features in the hash mapping. With different initialization of synaptic weights and different hash functions, capturing different dominant semantic information, can be learned with the bio-inspired learning rule. In this way, the data samples’ semantics are kept in the hash codes.
> We visualize the semantic capturing capabilities of bio-inspired hashing codes by comparing them to traditional hashing algorithms in Figure 3 in the paper. It is obvious that the cluster structure obtained by $\textit{SoftHash}$ is more compact compared to traditional $\textit{LSH}$ when using the same hash length. It indicates that the hash codes generated by $\textit{SoftHash}$ effectively capture more semantic information of data samples than those of traditional LSH.
>
> [1] Revisiting winner-take-all all (WTA) Hashing for Sparse Datasets
>
> R-Q3: As the same as most traditional LSH methods, bio-inspired hashing methods construct only a binary hash index for the nearest neighbor search.
> No other additional storage is required, and the binary form of the hash codes is very friendly to storage and search, bringing about an economic memory for semantic similarity computing.
> With respect to the index data structure, compared to traditional LSH methods with dense hash codes, bio-inspired methods use sparse hash codes to perform similarity search.
> In a hamming space, sparse vectors can be computed much faster than dense vectors.
> What's more, as introduced in the previous response, the winner-take-all mechanism ensures the output representations of bio-inspired hashing methods convey more semantic information.
> Therefore, with the above, bio-inspired hashing methods can implement fast retrieval/semantic computing.

---

### Official Review · Reviewer_GrZJ · 2023-10-24

**Soundness:** 2 fair
**Presentation:** 3 good
**Contribution:** 2 fair
**Rating:** 6
**Confidence:** 4

**Summary:**

The paper proposes a new data-dependent hashing algorithm called SoftHash based on a Hebbian-like update rule from biology. The method iteratively trains projection weights and biases using both the input data $x$ and output $y$, and generates sparse binary codes by a topk filter on $y$. Experiments show that SoftMax performs better than several previous methods in preserving the cosine similarity between the data points, as well as in dowsnstream similarity search tasks.

**Strengths:**

The presentation is in general clear and easy to follow. There are some grammar issues, e.g., "Such that, ...". A proofread is suggested.

In my understanding, the main algorithmic contribution is to improve the prior BioHash algorithm by replacing the hard thresholding with a softmax function. The idea seems plausible and should work, as one can always add a hyperparameter to softmax to make it a hard thresholding. The method introduces more flexibility.

**Weaknesses:**

1. Some similar contents have already appeared in other works. For example, the paragraph saying "$w_i$ is implicitly normalized" is similar to the text around Eq (3) in the BioHash paper (except that $Topk(y)$ is replaced by $y$). Eq (7) also has similar form in the BioHash paper. A clear comparison should be made in the paper regarding the difference and connections.

2. The design is not very well motivated and rather simple. If we write Eq (2) in your notation for SoftHash, then Oja's rule should be $\eta u_j(x_j-w_j^iu_j)$. The difference with Eq (3) is that the first $u_j$ is replaced by $y=softmax(u)$. The difference is that you added a non-linear softmax function on the scalar. Why is softmax used, can we use other functions? Is there also "biological interpretation" for that?

3. From the experiments, it seems that the SoftHash requires some delicate tuning on several floating parts. The choice of hyperparameters and tuning strategy need further explanation. Ablation study is also needed to understand the performance of the proposed method.
(1) SoftHash-1 and SoftHash-2 are not explained in the main paper, please revise. From the appendix, it seems that they refer to 2 different weight initialiation methods (uniform and Gaussian). Why is the performance gap so big given that this approach is learning-based? This is confusing to me. Also, why is the batch size set as 3584? It that carefully picked or just a random number?
(2) For BioHash, why you used $p=4$ and $\triangle=0.4$ and $0.3$? For SoftHash, how is the temperature paramter $T$ chosen? Did you tune these parameters?

**Questions:**

1. Does BioHash have the bias term $b$ in the model? How important is this bias term? Any empirical results for illustration?
2. I think "ConvHash" is usually referred to as "SimHash" or simply "LSH". Perhaps changing the name would be better.
3. Is the "Sokal and Michener similarity function" simply the Hamming similarity? If so, using the later is more straightforward.
4. In Table 2, what are the metrics when the code length is larger (256, 512)?
5. Is there an interpretation of Oja's rule in terms of gradient-based optimization? It seems that Eq (2) is the gradient of $||x-wy||^2$? Is it related to the clustering formualtion you mentioned?
6. There is a recent paper [SignRFF: Sign Random Fourier Features, NeurIPS 2023] that extends SimHash to non-linear features. It may perform better than SimHash on some datasets. Please consider adding this baseline as well as deep hashing methods to make the experiments stronger.

---

> ### Author Response · Authors · 2023-11-20
>
> Thank you for your thoughtful comments.
>
> R-W1&W2 Our algorithm design considers a lot about "winner-take-all" dilemma [1].
> Many winner-take-all hashing methods like BioHash may fall into the dilemma,
> where the semantic representation of the learned hash codes may be dominated by the majority class, leading to the semantics loss of the minor classes, which greatly limits its ability to catch the semantic information of the input data.
>
> In the energy function of BioHash (Eq.(10)), it is clear that each input sample belongs to only one neuron.
> It means only the output neuron with the largest value can be updated during the learning process, leading to the dilemma.
> For our hashing method, to alleviate this dilemma, we introduce a soft winner-take-all mechanism for learning the hash function.
> In contrast to the mentioned methods using a "hard" winner-take-all mechanism, our method makes non-winning classes not fully suppressed in the learning process.
> By assigning a credit score $\frac{e^{u_j+b_{j}}}{\sum_{l=1}^m e^{u_l+b_{l}}}$ in Eq (5) to each class, our method enables the minor classes to have a chance to influence the representative learning, making the learned hash codes improved with more semantic information. We have a detailed description of the connections and differences between SoftHash and BioHash in Appendix B.
>
> $\frac{e^{u_j+b_{j}}}{\sum_{l=1}^m e^{u_l+b_{l}}}$ in Eq (5) is in the form of softmax function.
> The reason why we use softmax here is to assign each class with a normalized credit score.
> Compared to other normalization functions, softmax is more general and smooth.
> The biological plausibility of "softmax" sources from the lateral inhibition between neurons, is mentioned in [2,3,4].
>
> [1] Domain Adaptive Semantic Segmentation without Source Data
>
> [2] SoftHebb: Bayesian inference in unsupervised Hebbian soft winner-take-all networks
>
> [3] Bayesian computation emerges in generic cortical microcircuits through spike-timing-dependent plasticity
>
> [4] Optimality of short-term synaptic plasticity in modeling certain dynamic environments
>
> R-W3:
> (1) SoftHash-1 and SoftHash-2 have different weight initialization methods, uniform initialization for SoftHash-1, and Gaussian initialization for SoftHash-2.
> The reason we believe that SoftHash-2 performs better is that Gaussian initialization is more effective in breaking symmetry compared to uniform initialization, leading to more representative feature combinations at the algorithm's beginning.
> In Gaussian initialization, weights are randomly sampled from a normal distribution, whereas in uniform initialization, weights are drawn from a uniform distribution.
> Gaussian initialization ensures that each neuron learns distinct features from the beginning, and improves the network's ability to capture input data characteristics during the learning process, enabling the learnable synaptic weights to generate more representative hash codes.
> We have added an ablation study in section 4.2 and descriptions of SoftHash-1 (SoftHash-2) in Table 2.
> We are sorry for the typo.
> In our study, the batch size is 1024 for all three datasets.
> It is determined by maximizing the usage of GPU memory.
> (2) For ease of comparison, we repeat the parameter setting of the previous work[1] for BioHash.
> For SoftHash, we have tuned the temperature parameter $T$ with \{5,10, 15, 20\} and set it to 10 according to its best performance.
>
> [1] Unsupervised Learning by Competing Hidden Units
>
> R-Q1: BioHash does not have the bias term.
> From the view of neuroscience, neuron activation occurs when the input signals reach a certain threshold, which is regarded as the neuron bias.
> Different neurons have different thresholds.
> Therefore, in order to precisely model the activities of neurons, the bias term should be seriously considered in the model.
> In practice, without the bias term, the learning algorithm may find it hard to fit the input samples, especially when the diversity of input features is large.
> As a result, the learned hash codes may lose effectiveness in finding similar data samples as shown in Table R-Q1.
>
> |            |        | Fashion-MNIST |        |        | CIFAR10 |        |        | CIFAR100 |        |
> | :--------: | :----: | :-----------: | :----: | :----: | :-----: | :----: | :----: | :------: | :----: |
> | Method     | 32     | 64            | 128    | 32     | 64      | 128    | 32     | 64       | 128    |
> | SoftHash_with_bias | 0.5080 | 0.6003        | 0.6712 | 0.2632 | 0.3570  | 0.4681 | 0.2872 | 0.4038   | 0.5163 |
> | SoftHash_without_bias | 0.5062 | 0.5952        | 0.6690 | 0.2217 | 0.2722  | 0.3104 | 0.2400 | 0.2899   | 0.3262 |

---

> ### Author Response · Authors · 2023-11-20
>
> R-Q2: We have changed "ConvHash" to "LSH" in the paper.
>
> R-Q3: "Sokal and Michener similarity function" is the Hamming similarity. We agree with you. We have made the modifications to the relevant content.
>
> R-Q4: In Table 2, we use mean average precision (MAP) for evaluation when the code length is larger. In the image retrieval task, we use MAP to measure the hamming ranking quality of various code lengths.
>
> R-Q5: Your observation is keen, which enlightens us a lot.
> Next, we will try to explain Oja's rule from the view of gradient descent.
> First, let us look at the following equation
> \begin{equation}
>   \frac{\partial\|(x_i - w_{j}^iy_j)\|^2}{\partial w_j^i } = -2y_j (x_i - w_{j}^iy_j)
> \end{equation}
> It is interesting to find that the gradient of $\|(x_i - w_{j}^iy_j)\|^2$ is the same as the Oja rule (Eq (2)).
>
> The $w_{j}^{i}$ in the numerator is just a regularization term that controls $w_{j}^{i}$ not to grow too big.
> If we ignore this term, the update rule of Oja can be rewritten to minimize the loss function as
> \begin{equation}
> \underset{w_j^i}{\text{minimize}} \sum_{i=1}^d \|(x_i-y_j)\|^2
> \end{equation}
> where $y_j=\sum_i x_i w_{j}^i$.
> It is easy to see the loss function expects the output neuron ($y_j$) can act similarly as most of the input neurons ($x_i$s) as possible.
> Therefore, at the end of this unsupervised learning, $y_j$ will be determined by the most dominant correlated $x_i$s.
> In other words, the input samples with similar values on the dominant semantic features will have similar hashing values and be hashed into the same bucket.
>
> This finding is very interesting.
>
> We will add it to our final paper.
>
> Thanks for your inspiration.
>
> R-Q6: We have added SignRFF as an important reference in our section of related work. Next, we are going to do our best to implement SignRFF as a comparative baseline of our experiment in our final version.
>
> Thank you again for your valuable feedback.

---

> ### Comment · Reviewer_GrZJ · 2023-11-23
>
> Thanks for the reply. Some background and motivation are more clear to me now. I'm glad that my comments are helpful to improve your work.
>
> W3: If symmetry might be the key factor here, why not using a symmstric uniform distribution as initialization? In practice Gaussian is more common, so proposing the (positive-only) uniform initialization may not offer much contribution or insights here.
>
> Q4: What I meant is that I'm curious about the comparisons when the embedding dimension is larger. Now you only presented the results up to k=128. How about k=256, 512, 1024 etc? Longer codes are useful when high precision and recall are required. Please consider adding them to make the results more complete.

---

> ### Author Response · Authors · 2023-11-23
>
> Thank you for your kindly reply.
>
> R-W3: We agree with your point. To fully study if symmetry is the key factor, we will add a result of the initialization method of the symmetric uniform distribution, which has the same scale range as the Gaussian distribution, in the final paper.
>
> R-Q4: You are right. In our work, longer codes (k=256, 512) show better retrieval performance on three datasets (Appendix D.2). We will add k=1024 to make the results more complete in the final version.

---

### Official Review · Reviewer_xpXr · 2023-10-27

**Soundness:** 3 good
**Presentation:** 3 good
**Contribution:** 3 good
**Rating:** 5
**Confidence:** 3

**Summary:**

This paper introduced a SoftHash, which is a data-dependent hash algorithm. It produces sparse and high-dimensional hash codes. Also, unlike other algorithms such as BioHash, this paper did not use a hard winner-take-all(wta) mechanism. They adopted a soft wta mechanism with an exponential function. The authors conducted experiments on similar image searches and semantical similar word searches.

**Strengths:**

This paper proposed a new high-dimensional hashing algorithm. The authors use soft WTA to enable the learning of weak-correlated data, which differs from existing work.

 The paper demonstrated the differences between their algorithm and one existing algorithm BioHash.

The explanation of SoftHash is clear.

**Weaknesses:**

The experiment part was not clear to me. Many important details are in the appendix or are missing, such as the train/test set split.  I keep the questions in the next section.

**Questions:**

1. Appendix D.1 mentioned that for test: "10 query per class for cifar100, 100 for cifar10".  The test size seems small to me. In BioHash, they used 1000 queries per class for cifar10. Any reason why you chose a smaller query size?
2. Appendix D.1 mentioned that "Ground truth is the top 1000 nearest neighbors of a query in the database, based on Euclidean distance between pairs of images in pixel space. " whereas, in BioHash, the ground truth is based on class labels. For me, it makes more sense to use class labels. Is there any reason for using Euclidean distance?
3. Appendix D.1 mentioned that "Output dim is 2000". It is smaller than the 3072-dim input in cifar10/cifar100. Can you explain if this still satisfies high-dimensional hash?
4.  SoftHash-1 and SoftHash-2 used different parameter initialization. The conclusion is that SoftHash2 is better than SoftHash1. But any intuition as to why it's better? Instead of putting them in the main result table, you can put different initialization results as an ablation study. It's really confusing when looking at the main result table because there is no definition of SoftHash 1 in the main paper.
5. I couldn't find train/test split and output dimensions for word search experiments.
6. The reference for "Can a fruit fly learn word embeddings?" needs to be updated: not arxiv preprint; it was published at iclr 2021.

---

> ### Author Response · Authors · 2023-11-20
>
> Thank you for your thoughtful comments, which prompt us to improve our manuscript. We would like to address your concerns and answer your questions in the following.
>
> R-W1: We have added the description of the train/test set split in the $\textit{Datasets}$ part of section 4.1 and rewritten the experiment settings of section 4.1 and section 5.1 in $\textcolor{blue}{BLUE}$ color.
>
> R-Q1: To further illustrate the effectiveness of our algorithm, in Table R-Q1, we add the experimental results of "10,000 images with 10 classes". It shows that when the query size becomes large, SoftHash still outperforms other hashing methods with different hash lengths.
>
>  Table R-Q1
> |          |        | Fashion-MNIST |        |        | CIFAR10 |        |        | CIFAR100 |        |
> | :------: | :----: | :-----------: | :----: | :----: | :-----: | :----: | :----: | :------: | :----: |
> | Method   | 32     | 64            | 128    | 32     | 64      | 128    | 32     | 64       | 128    |
> | SH       | 0.3492 | 0.4301        | 0.4743 | 0.1369 | 0.1518  | 0.1905 | 0.1136 | 0.1569   | 0.1962 |
> | ITQ      | 0.3513 | 0.4305        | 0.4949 | 0.1901 | 0.2337  | 0.2685 | 0.1963 | 0.2418   | 0.2769 |
> | BioHash  | 0.4830 | 0.5270        | 0.5529 | 0.2139 | 0.2806  | 0.3289 | 0.2422 | 0.3066   | 0.3501 |
> | SoftHash | 0.5205 | 0.6072        | 0.6786 | 0.2791 | 0.3748  | 0.4840 | 0.3021 | 0.4003   | 0.5064 |
>
> R-Q2: In this paper, we focus on designing a data-dependent hashing algorithm $\textit{SoftHash}$ to encode dense continuous features into sparse binary codes for fast searching, meanwhile, adapting semantic knowledge from input data features to output hash codes for semantic clustering.
> Considering the above, we do not choose class-label-based evaluation, which is a bit coarse to reflect the effect of semantic clustering of our algorithm. On the other hand, Euclidean-distance-based evaluation is selected, which is more soft and refined to assess the semantic representativeness of output hash codes. Moreover, some datasets like Glove naturally have no class labels, which can not be conducted with class-label-based evaluation.
>
> R-Q3: The reason we set the output dim as 2000 is motivated by the fact that in the factory circuit of Fruit Fly there are approximately 2000 Kenyon cells for information encoding [1]. Considering this biological basis,  2000 is used as the output dim in the experiments.  "2000 is high" does not mean that the output dimension is higher compared to the input dimension. Here, we say "2000 is high" which means that compared to the traditional LSH methods like [2,3] hashing data into dense low-dimensional(d < 2000) space, our method's 2000 output dimension is a higher number. The main objective of the experiments is to show the superiority of our sparse high-dimensional hash representation over other dense low-dimensional hash representations.
>
> [1] A neural algorithm for a fundamental computing problem; [2] Spectral hashing; [3] Iterative quantization: A procrustean approach to learning binary codes for large-scale image retrieval.
>
> R-Q4: SoftHash-1 and SoftHash-2 have different weight initialization methods, uniform initialization for SoftHash-1, and Gaussian initialization for SoftHash-2. The reason we believe that SoftHash-2 performs better is that Gaussian initialization is more effective in breaking symmetry compared to uniform initialization, leading to more representative feature combinations at the algorithm's beginning.
> In Gaussian initialization, weights are randomly sampled from a normal distribution, whereas in uniform initialization, weights are drawn from a uniform distribution. Gaussian initialization ensures that each neuron learns distinct features from the beginning, and improves the network's ability to capture input data characteristics during the learning process, enabling the learnable synaptic weights to generate more representative hash codes. We have added an ablation study in section 4.2 and descriptions of SoftHash-1 (SoftHash-2) in Table 2.
>
> R-Q5: Thank you for your comments. We have rewritten the description of the experiment settings in Section 4.1 and Section 5.1, including train/test split in the $\textit{Datasets}$ part of Section 4.1. The output dimensions for the word search experiments were set to 2,000. It has been added in Appendix E.1.
>
> R-Q6: Thank you for your correction. We have made the modifications in the references.

---

### Official Review · Reviewer_qurX · 2023-11-01

**Soundness:** 3 good
**Presentation:** 3 good
**Contribution:** 3 good
**Rating:** 6
**Confidence:** 3

**Summary:**

The paper proposes SoftHash, a data-dependent hashing algorithm. To overcome the randomness of LSH, SoftHash learns the hashing projection function using a Hebbian-like learning rule coupled with the idea of Winner-Take-All (WTA). This allows SoftHash to adapt to the input data manifold and generate more representative hash codes. The authors also introduce a soft WTA rule, whereby the non-winning neurons are not fully suppressed in the learning process. This allows weakly correlated data to have a chance to be learned, which further improves the semantic representation capability of SoftHash.

**Strengths:**

Here are some specific strengths of the paper:

1. Overall, the paper is well-written and presents a novel and effective hashing algorithm. The authors provide a clear motivation for their work, and their experimental results are convincing.

2. The authors evaluate SoftHash on some real-world datasets and tasks, and their experimental results demonstrate that SoftHash significantly outperforms some baseline methods.

**Weaknesses:**

1. It would be better to include more baselines including Mongoose paper's learnable hash functions[1].

2. Maybe the authors could justify more on the theoretical analysis of the motivation of Softhash. Similar studies would be [2]

[1] MONGOOSE: A Learnable LSH Framework for Efficient Neural Network Training
[2] Learning to Hash Robustly, Guaranteed

**Questions:**

1. How to provide search quality guarantees for Softhash in retrieval?

2. What is the convergence rate of the SoftHash learning process?

---

> ### Author Response · Authors · 2023-11-20
>
> Thank you for your valuable comments!
>
> R-W1: Thank you for pointing out this excellent reference. We are in the process of reimplementing MONGOOSE and testing it on the evaluation benchmarks we have used. We will add the experimental results to our paper once it is available.
>
> R-W2 & Q1: Thank you for your comments. The key idea of SoftHash originates from Drosophila’s olfactory circuit [3], which utilizes sparse spike codes to convey the information of dense sensory signals. In particular, Hebbian-like rules are often used to learn such sparse codes. In this paper, our Hebbian-like learning rule is designed as $y_jx_i-y_ju_jw_j^i$, which works similarly to the Oja rule [4]. If the input and output are highly correlated, the first term will enhance the connection weight between the input and output neuron; The second term will give a punishment update when the weight is already large enough. The item $y_j$ is computed according to the Bayesian posterior probability, which allows every possible hash class to be able to influence the update at each iteration. In this way, our hash function can be trained in a soft manner.
>
> Unlike randomized hash functions, the theoretical bounds for learning-based hashing methods are often lacking [5]. Although recently [2] proves a correctness bound for its tree-learning-based hashing method, its theoretical analysis cannot be easily applied to other learning-based methods, such as neural-network-based hashing methods. The reason is that [2] is a white-box algorithm, in which independence and randomness are introduced by the authors’ delicate design. With the well-designed randomness setting, the theoretical bound can be achieved as expected. However, neural-network-based hashing methods like ours are black-box algorithms, in which the function weight may converge to any saddle point with unknown probability. Therefore, it is challenging to provide a theoretical guarantee for our method. However, from the view of machine learning, our algorithm can be treated as an energy-based model as described in Equation (7) in the paper. We have proved that our learning rule can consistently reduce the energy value in Appendix A, which ensures the convergence of our method. Furthermore, we also provided an analysis on the time complexity of the algorithm.
>
> [1] MONGOOSE: A Learnable LSH Framework for Efficient Neural Network Training
> [2] Learning to Hash Robustly, Guaranteed
> [3] A neural algorithm for a fundamental computing problem
> [4] A Simplified Neuron Model as a Principal Component Analyzer
> [5] Learning to Hash for Indexing Big Data—A Survey
>
> R-Q2: Thank you for your comment. In our experiments, most of the algorithms can converge to a good result within 20 epochs.

---

### Author Response · Authors · 2023-11-21
**General Response**

We thank all reviewers for their time and helpful comments. We have responded to all reviewers' comments and uploaded a revised version of our manuscript with all changes marked in blue.

In this study, we consider a lot about the "winner-take-all" dilemma, many winner-take-all hashing methods like BioHash may fall into the dilemma. This dilemma occurs when the semantic representation within the learned hash codes is influenced by the majority class. Consequently, it results in the loss of semantic information from minor classes, limiting the model's capability to capture semantic information from the input data.
To alleviate this dilemma, our method makes non-winning classes not fully suppressed in the learning process by assigning a credit score to each neuron. It enables the minor classes to have a chance to influence representative learning, enriching the learned hash codes with more comprehensive semantic information.

We have a detailed description of the connections and differences between SoftHash and BioHash in Appendix B. We revised the experiment settings in section 4.1. We have added an ablation study in section 4.2 and descriptions of SoftHash-1 (SoftHash-2) in Table 2.
Additionally, we have added graph hashing SGH as the baseline, doing our best to strengthen the experiment with more baselines including MONGOOSE and SignRFF in our final version.

In summary, we greatly appreciate the professional comments provided by the reviewers, which inspired us a lot. We will adopt their thoughtful opinions to make our final paper more enriched and convincing. Additionally, to further provide more practical references to this paper, we will upload and open the code in the final.

Thanks,

Paper 2335 Authors

---

### Meta-Review · Area_Chair_dUEw · 2023-12-08

**Metareview:**

Thanks for your submission to ICLR.

This paper proposed a novel hashing scheme called SoftHash, which is biologically motivated.  The authors develop the technique, and then compare it with other data-dependent hashing schemes for nearest neighbor search tasks.

This is very much a borderline paper, and the reviewers are on the fence with it.  On the positive side, the proposed approach is simple and novel, and on the data sets shown, performs well.  On the negative side, the reviewers noted that there are issues of clarity, some issues with experimental results, a need to perform tuning, and a lack of some baselines.  The rebuttal helped somewhat, as one of the more negative reviewers indicated that they were leaning weak accept.  That still left the paper very borderline.

I took a close look at this paper.  I tend to agree with some of the criticisms of the paper.  To be clear, I think that this could be a very nice paper, and I think it's worth continuing to work on it to make it stronger.  But I really think it needs an additional round of editing and updating before it's ready for publication.  In particular, the paper falls in a very crowded area (ML-based hashing techniques) that has been consistently studied over the last 20 or so years, so the bar is set pretty high for new papers in this space.  Most successful new papers in this area either take a theoretical view (e.g., designing hashing algorithms with provable guarantees) or have extremely strong empirical results (or both).  If you look at a recent survey such as "Hashing Techniques: Survey and Taxonomy", there are about 200 references of various hashing-based methods, with several of them falling under the category of data-dependent hashing techniques (e.g., product quantization, AGH, AQBC, Inductive Manifold Hashing, and others).  I would have expected a much more in-depth discussion and comparison to some of these baselines.  Instead, I found the experiments fairly weak, with both missing baselines as well as somewhat limited results (fairly small data sets that don't really require hashing at all).  Furthermore, I was a bit underwhelmed by the response to one of the reviewer criticisms about missing baselines, which did not follow up with results on the suggested baseline.  So I think the paper could use some significant strengthening in this area.

**Justification For Why Not Higher Score:**

The reviews were borderline, leaning somewhat to accept, but there are several key issues with the experimental results, and the author rebuttal was somewhat underwhelming.

**Justification For Why Not Lower Score:**

N/A

---

### Decision · Program_Chairs · 2024-01-16

Reject